# Toward Understanding the Transferability of Adversarial Suffixes in Large Language Models

**Sarah Ball**[*]                                          *sarah.ball@stat.uni-muenchen.de*
*LMU Munich*
*Munich Center for Machine Learning (MCML)*

**Niki Hasrati**[*]                                                *nhasrati@andrew.cmu.edu*
*Carnegie Mellon University*

**Alexander Robey**
*Carnegie Mellon University*

**Avi Schwarzschild**
*Carnegie Mellon University*

**Frauke Kreuter**
*LMU Munich*
*Munich Center for Machine Learning (MCML)*
*JPSM University of Maryland*

**J. Zico Kolter**
*Carnegie Mellon University*

**Andrej Risteski**
*Carnegie Mellon University*

**Reviewed on OpenReview:** *https://openreview.net/forum?id=wQZmcEZCUK*

## Abstract

Discrete optimization-based jailbreaking attacks on large language models aim to generate short, nonsensical suffixes that, when appended onto input prompts, elicit disallowed content. Notably, these suffixes are often *transferable*—succeeding on prompts and models for which they were never optimized. And yet, despite the fact that transferability is surprising and empirically well-established, the field lacks a rigorous analysis of when and why transfer occurs. To fill this gap, we identify three statistical properties that strongly correlate with transfer success across numerous experimental settings: (1) how much a prompt without a suffix activates a model's internal refusal direction, (2) how strongly a suffix induces a push away from this direction, and (3) how large these shifts are in directions orthogonal to refusal. On the other hand, we find that prompt semantic similarity only weakly correlates with transfer success. These findings lead to a more fine-grained understanding of transferability, which we use in interventional experiments to showcase how our statistical analysis can translate into practical improvements in attack success. ⭘

---

[*]Equal contribution.
LLM use: Claude Sonnet 4, GPT-4, and GPT-5 for editing and coding assistance (Anthropic, 2025; OpenAI, 2024; 2025).

# 1 Introduction

Adversarial examples—carefully crafted input perturbations that can make models behave in undesirable ways—remain a fundamental obstacle to achieving robustness across deep learning tasks and data modalities (Carlini & Wagner, 2017; Goodfellow et al., 2015; Madry et al., 2018). A particularly puzzling property of these perturbations is their *transferability*—perturbations optimized for one input or model are often effective on others (Papernot et al., 2016; Szegedy et al., 2014).

Although initially discovered in the context of image classification (see, e.g., Salman et al. (2020); Tramèr et al. (2017)), transferability has resurfaced as a key aspect of *jailbreaking* large language models (LLMs) to elicit harmful responses (Wei et al., 2023). While jailbreaks are typically optimized for a particular model and input prompt, recent empirical findings conclusively show that jailbreaks often transfer between models, despite differing architectures and training data (Andriushchenko et al., 2025; Chao et al., 2025). Of particular note are discrete optimization-based jailbreaking algorithms that generate short, nonsensical suffixes that, when appended onto a prompt requesting harmful content, return a compliant response (Geisler et al., 2024; Wallace et al., 2019; Zou et al., 2023). And while the transferability of suffix-based attacks is empirically well-established, the field lacks a fine-grained understanding of when, why, and to what extent transfer occurs for these attacks.

In this paper, we identify features that are predictive of suffix-based transfer success by conducting a statistical and interventional study of the following questions: (1) Why are some prompts more susceptible to suffix-based attacks than others; (2) Which properties of a given suffix lead to successful transfer; and (3) What internal model mechanisms govern transfer success? Our study of these questions includes analysis of *intra-model transfer*—generalization across prompts within the same model—and *inter-model transfer*—generalization across models with the same prompt. Our main findings, which rely on notions related to *refusal directions* (Arditi et al., 2024), are as follows:

- **Prompt refusal connection:** Prompts corresponding to activations that are less aligned with a model's refusal direction are easier to successfully jailbreak, leading to more transfer.
- **Suffix push and orthogonal shift:** Suffixes that successfully transfer are primarily characterized by a strong antiparallel shift away from a model's refusal direction; perturbations orthogonal to this direction play a secondary and model-dependent role.
- **Prompt semantic similarity.** Prompt semantic similarity only weakly predicts transfer, which suggests that the geometry of suffix activation spaces is only loosely tied to linguistic form.

While variants of these quantities have appeared in prior work, we conduct a large-scale statistical and interventional analysis involving the optimization of 10,000 adversarial suffixes per model to rigorously quantify their effect on transferability. Moreover, we introduce algorithmic interventions that improve the success rates of existing attacks; we hope that this analysis informs the design of future defenses.

# 2 Related work

**Transferability of adversarial examples.** Over the past decade, the transferability of adversarial attacks has been observed across data modalities, architectures, and training schemes (Carlini & Wagner, 2018; Goodfellow et al., 2015; Neekhara et al., 2019; Ren et al., 2019; Taori et al., 2019). This finding has prompted various theories that seek to diagnose when and why transferability succeeds, particularly in the context of computer vision. While Tramèr et al. (2017) identify distributional conditions that lead to transfer in linear and quadratic models, Demontis et al. (2019) contend that other factors, including model complexity and gradient similarity, influence transferability. On the other hand, Ilyas et al. (2019) find that different models tend to learn similar non-robust features, making them susceptible to transfer attacks. In contrast to existing research, we provide a statistical and interventional study, which (a) concerns language, rather than images, and (b) identifies distinct features behind transferability based on a mechanistic interpretability analysis of activation spaces (Arditi et al., 2024).

**Transferability of jailbreaks.** The discovery that many distinct jailbreak strategies induce transfer across LLMs has renewed interest in model security (Jain et al., 2023; Robey et al., 2025; Zou et al., 2024). While

these varied attack modalities have helped identify model blind spots, this diversity also complicates the task of identifying the principles underlying the success of transferability. To this end, we focus on *suffix-based* jailbreaks (Jones et al., 2023; Liu et al., 2024; Zhu et al., 2024), since they admit structure that facilitates decoupling the effect of the prompt and the suffix. Because attacks from this family are all structurally similar, in this paper, we focus on the most frequently used, well-studied variant: Greedy Coordinate Gradient (GCG) (Zou et al., 2023).

**Mechanistic analyses of model safety.** Our results focus on a mechanistic analysis of jailbreak transferability, building on previous works that give a mechanistic interpretation of model safety. Most relevant is the work of Arditi et al. (2024), who identify a "refusal vector"—a direction in activation space that, when subtracted, reduces refusal on harmful prompts and, when added, triggers refusal on harmless ones. Follow-up studies further demonstrate that different jailbreak strategies alter the model's internal representation of harmfulness in distinct ways (Ball et al., 2026), often making harmful prompts appear more similar to benign prompts (Jain et al., 2024; Lin et al., 2024). By contrast, in this paper, we offer statistical and interventional analyses of the mechanisms behind transferability, which lead to a finer-grained understanding of when and why transfer succeeds.

## 3 Setting the stage: definitions and features

We next define preliminary quantities used throughout the paper, and formally define features of prompts and suffix that we analyze in this paper.

### 3.1 Preliminaries

We consider two forms of transfer. *Intra-model transfer* measures whether an adversarial suffix $s$, optimized for a particular prompt $p$, also succeeds when applied to different prompts $p'$ on the same model. *Inter-model transfer* measures whether an adversarial suffix $s$, optimized for a particular prompt $p$ and model $m$, also succeeds on a different model $m'$—either on the same prompt $p$ or a new prompt $p'$. We refer to the prompt used to optimize or generate a suffix as the *source prompt*, and the prompt on which that suffix is evaluated as the *target prompt*. To measure these properties, we also define the following:

**Definition 1 (Attack success rate (ASR))** *Given a suffix $s$, let $n^s_{jailbroken}$ denote the number of prompts for which appending $s$ results in a jailbroken response, and let $n^s_{total}$ denote the total number of prompts tested with suffix $s$. We define the attack success rate (ASR) as:* $ASR(s) := \frac{n^s_{jailbroken}}{n^s_{total}}$.

**Definition 2 (Refusal direction (Arditi et al., 2024))** *Given a set containing harmful and harmless prompts, let $\mathbf{a}^{i,\ell}_{harm}$ and $\mathbf{a}^{j,\ell}_{harmless}$ denote residual stream activation vectors for the final token at layer $\ell$ for the i-th harmful prompt and the j-th harmless prompt, respectively. The* **refusal direction** $\mathbf{v}^l_{refusal}$ *at layer $\ell$ is defined as the difference between the average activations among the prompts, namely*

$$\mathbf{v}^\ell_{refusal} = \left(\frac{1}{n}\sum_{i=1}^n \mathbf{a}^{i,\ell}_{harm}\right) - \left(\frac{1}{m}\sum_{j=1}^m \mathbf{a}^{j,\ell}_{harmless}\right).$$

The refusal direction compares the activations of contrastive pairs of harmful and harmless prompts in order to extract a single vector in representation space that captures the model's internal representation of harmfulness. Consistent with Arditi et al. (2024), we extract the refusal direction at the *optimal layer* (see Appendix A for details). Thus, for brevity, we often do not include the layer index.

### 3.2 Introducing the features

Our aim is to study features of prompts and suffixes that correlate with successful transfer. Several of the features we consider are related to the geometry of LLM activation spaces via the so-called *refusal direction* (see Definition 2)—a direction in activation space that triggers refusal when added to harmless prompts and suppresses refusal when subtracted from harmful prompts (Arditi et al., 2024). Before formally defining each quantity in §3.3, we first informally define each quantity of interest.

1. **Semantic similarity of prompts** (Definition 3). Does a suffix $s$ optimized for a prompt $p$ transfer more reliably to another prompt $p'$ when their representations are similar?

2. **Refusal connectivity of the prompt** (Definition 4). Are some prompts more aligned with the refusal direction (e.g., prompts related to concepts emphasized in model alignment), and are prompts aligned with the refusal direction less susceptible to transfer?

3. **Suffix push** (Definition 5). Are suffixes that induce a larger shift in the opposite (antiparallel) direction from the model's refusal direction more likely to transfer?

4. **Orthogonal shift of the suffix** (Definition 6). Are suffixes that induce a larger shift orthogonal to the model's refusal direction more likely to transfer?

Following the large body of work evincing the existence of a refusal direction in various models, the latter three definitions correspond to the following intuitive hypotheses: (a) prompts aligned with the refusal direction are less likely to transfer, (b) suffixes that induce an antiparallel shift are more likely to transfer, and (c) prompts that induce an orthogonal shift are more likely to transfer. In §3.3, we formally define these quantities, which will serve as the central objects of study in §5.

### 3.3 Formal definitions

We next formalize the quantities informally introduced in §3.2. Note that all activations are extracted at the same layer as the refusal direction (see Appendix A for details).

**Definition 3 (Semantic similarity)** *The semantic similarity $sim_{pp'}$ of two prompts $p$ and $p'$ is defined as the cosine similarity of some chosen embeddings $E(p)$ and $E(p')$, namely*

$$sim_{pp'} := \frac{\langle E(p), E(p') \rangle}{\|E(p)\| \cdot \|E(p')\|}.$$

We calculate these embeddings in two different ways—with activations from the model itself and with embeddings extracted from the sentence embedding model "all-mpnet-base-v2" (UKPLab, 2025).

**Definition 4 (Refusal connectivity)** *Let $\mathbf{a}_i^{base}$ denote the residual stream activation vector at the end-of-instruction token for the $i$-th harmful prompt. Given a refusal direction $\mathbf{v}_{refusal}$ (as defined in Arditi et al. (2024)), the* refusal connectivity *is measured via the quantities*

$$s_i^{base} := \langle \mathbf{a}_i^{base}, \mathbf{v}_{refusal} \rangle \qquad and \qquad cos(\mathbf{a}_i^{base}, \mathbf{v}_{refusal}) = \frac{\langle \mathbf{a}_i^{base}, \mathbf{v}_{refusal} \rangle}{\|\mathbf{a}_i^{base}\| \cdot \|\mathbf{v}_{refusal}\|}.$$

**Definition 5 (Suffix push)** *Let $a_{ij}^{suffix}$ denote the activations for the string $\langle p_i, s_j \rangle$, which represents the concatenation of prompt $i$ with suffix $j$. For a prompt-suffix pair $(i, j)$, the* suffix push *quantifies the change in refusal connectivity when adding a suffix to the prompt, namely*

$$\Delta_{ij}^{push} := \langle \mathbf{a}_i^{base}, \mathbf{v}_{refusal} \rangle - \langle \mathbf{a}_{ij}^{suffix}, \mathbf{v}_{refusal} \rangle.$$

**Definition 6 (Orthogonal shift)** *Let the projection of an activation vector $\mathbf{a}$ onto the refusal direction $\mathbf{v}_{refusal}$ be defined as $\mathbf{p}(\mathbf{a}) := \frac{\langle \mathbf{a}, \mathbf{v}_{refusal} \rangle}{\|\mathbf{v}_{refusal}\|^2} \cdot \mathbf{v}_{refusal}$. The* orthogonal shift *for a prompt-suffix pair $(i, j)$ measures the change in activations perpendicular to the refusal direction, namely*

$$\delta_{ij}^{\perp} := \left\| \left( \mathbf{a}_{ij}^{suffix} - \mathbf{p}(\mathbf{a}_{ij}^{suffix}) \right) - \left( \mathbf{a}_i^{base} - \mathbf{p}(\mathbf{a}_i^{base}) \right) \right\|_2.$$

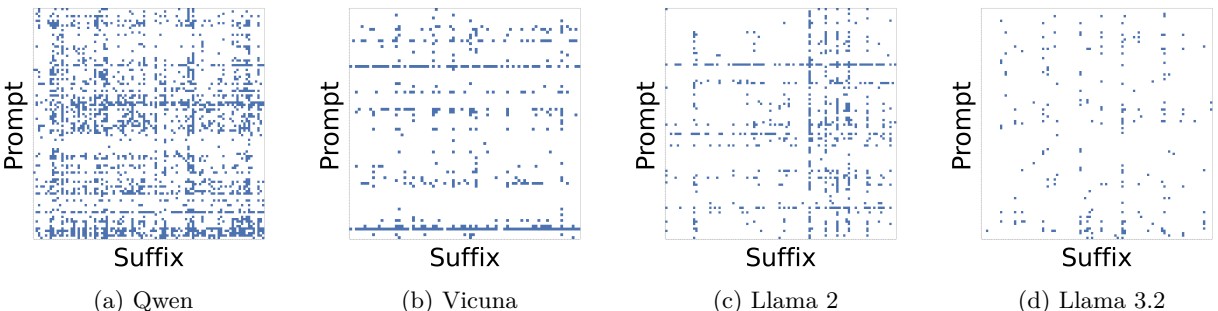

|  |  |  |  |
| :---: | :---: | :---: | :---: |
| (a) Qwen | (b) Vicuna | (c) Llama 2 | (d) Llama 3.2 |

Figure 1: Intra-model transfer with one suffix sampled uniformly at random from the 100 suffixes generated for each source prompt using different random seeds. Rows and columns are restricted to prompts that are not jailbroken without a suffix. A cell at column $x$ and row $y$ is colored when the sampled suffix generated for source prompt $x$, when appended to target prompt $y$, successfully jailbreaks the model.

## 4 Experimental setup

This section details the selection of models, the dataset of harmful prompts, the procedure for generating adversarial suffixes, and the approach for evaluating their jailbreaking success.

**Models.** We use Qwen-2.5-3B-Instruct (Qwen et al., 2025), Llama-3.2-1B-Instruct (Meta AI, 2024), Vicuna-13B-v1.5 (Chiang et al., 2023), and Llama-2-7B-Chat (Touvron et al., 2023). While these models are all safety-trained, this list includes models considered easy to jailbreak (e.g., Vicuna) and harder to jailbreak (e.g., Llama-2). This diversity is crucial for assessing the generalizability of our findings across models with different architectures and safety alignment characteristics. A table highlighting relevant aspects of these models is included in Appendix B.

**Data.** We use the JailbreakBench dataset (Chao et al., 2024), which contains 100 harmful questions and answer targets on topics spanning various risk categories as defined by OpenAI's usage policies.

**Generation of adversarial suffixes.** We generate suffixes for each JailbreakBench prompt (Chao et al., 2024) using the GCG algorithm (Zou et al., 2023). To obtain stable measurements of the statistical quantities outlined above, we generate 100 distinct suffixes per prompt (i.e., 10,000 suffixes per model) by varying GCG's random seed.

**Evaluating jailbreak success.** To evaluate whether jailbreaks succeed, we use a Llama-3-Instruct-70B judge with the system prompt from JailbreakBench following the recommendation of (Chao et al., 2024, Table 1), who evaluated the effectiveness of six commonly used jailbreaking judges. To additionally evaluate the reliability of the LLM judge in our setting, we conducted a human validation experiment (see Appendix B), revealing an agreement rate of 98.65%.

## 5 Analysis of the factors correlated with transfer

Toward understanding the effect of each quantity introduced in §3.2, we first record basic transfer statistics (§5.1). We next qualitatively and quantitatively analyze each quantity (§5.2, §5.3, and §5.4). We then provide a joint statistical analysis to estimate the *predictive strength* of the factors in relation to each other (§5.5). We conclude with an exploration of how these insights can be used to produce more transferable suffixes.

### 5.1 Qualitative analysis of transfer statistics

As a preliminary step, we highlight some illustrative properties of transfer that can be gleaned from the raw statistics of suffix-based transfer. We focus on two main scenarios: intra-model transfer (Figure 1) and inter-model transfer (Figure 2). Figure 6 in Appendix C additionally visualizes jailbreak success across multiple random initializations for all four models. Figures 1, 2, and 6 restrict their prompt axes to prompts

that are not jailbroken without a suffix. For visual comparability across models, in Figure 1, each source prompt is represented by one suffix sampled from the 100 suffixes generated using different random seeds; however, our quantitative analyses use the full set of generated suffixes. As in §3.1, source prompts are the prompts used to generate suffixes, while target prompts are the prompts on which those suffixes are evaluated.

**Model susceptibility to jailbreaking.** Figure 1 reveals that models exhibit different susceptibilities to adversarial suffixes. Specifically, Qwen shows the densest pattern of successful intra-model transfer, while Llama 3.2 is substantially sparser. Vicuna and Llama 2 fall between these extremes, with transfer concentrated around particular prompts and suffixes rather than uniformly across the matrix. Figure 6 in Appendix C shows that the model-level differences in jailbreak susceptibility are not driven solely by the single sampled suffix used in Figure 1.

**Intra-model transferability.** Within individual models, the success of adversarial suffixes is not uniform. Figure 1 highlights that certain prompts are consistently more vulnerable; these appear as horizontal bands with a higher density of successful jailbreaks. Conversely, some adversarial suffixes exhibit greater potency, successfully compromising a larger set of prompts within the same model. These are identifiable as denser vertical bands in Figure 1. A noteworthy phenomenon is the off-target efficacy of some suffixes: a suffix optimized for a specific prompt (i.e. its corresponding diagonal entry in Figure 1) may fail to jailbreak its prompt but successfully jailbreak other prompts (off-diagonal) within the same model.

**Inter-model transferability.** Suffixes also transfer across models (Figure 2). Using suffixes sampled from the same pool of suffixes generated with multiple random initializations, we observe an asymmetry: suffixes optimized on a more aligned model (Llama 3.2) transfer better to a less aligned one (Qwen) than vice versa.

**Takeaways.** In sum, transfer occurs within and across models, but success depends on the model, the prompt's vulnerability, and the potency of the suffix. The next sections analyze these factors.

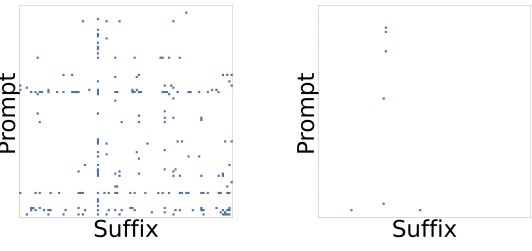

(a) Llama 3.2 to Qwen     (b) Qwen to Llama 3.2

Figure 2: Inter-model transfer between Llama 3.2 and Qwen. In each subfigure, column $x$ is a suffix generated on the source model and row $y$ is a target-model prompt; a colored cell means appending suffix $x$ to prompt $y$ successfully jailbreaks the target model. Rows are restricted to target prompts that are not jailbroken by the target model without a suffix, and columns are restricted to suffixes whose source prompts are not jailbroken by the source model without a suffix.

### 5.2 Semantic similarity

As outlined in §3.2, we aim to determine whether the semantic similarity between the embeddings of two prompts $p$ and $p'$ is predictive of the transferability of a suffix originally optimized for $p$.

**Statistical analysis setup.** We set up a quantitative framework for estimating the effect of semantic similarity ($\text{sim}_{pp'}$, Definition 3) on transferability. Every observation is a single (source prompt $p$, suffix $s$, target prompt $p'$) triple with binary outcome $y_{psp'} \in \{0, 1\}$ indicating whether $p'$ was jailbroken by $s$, predicted from the feature vector $\mathbf{x}_{pp'} := [1, \cos(E(p), E(p'))]$. Same-prompt pairs ($p = p'$) are excluded, giving $N = 100 \times P(P - 1) = 990{,}000$ observations per model. We fit a generalized linear mixed model (GLMM) with a logistic link on $\mathbf{x}_{pp'}$, standardizing the cosine similarity term to have mean 0 and variance 1. To account for the non-independence of observations sharing the same prompts or suffix, we include random intercepts for each grouping level: random intercepts for target prompt and for suffix nested within source prompt[1] capturing the additional variance shared by the 100 suffixes co-optimized on the same source prompt. All models are fit using the `lme4` package in R (Bates et al., 2015).

Table 1 shows the resulting regression coefficients, indicating a statistically significant and positive relationship for all models. Following the standard statistical rules-of-thumb (Chen et al., 2010), we conclude that most of the effect sizes are small. Consistent with this, the marginal $R^2$ values ($R^2_m = 0.000$–$0.048$) indicate that

---

[1]in `lme4` notation (Bates et al., 2015): `(1 | target_prompt_id) + (1 | source_prompt_id / suffix_id)`

Table 1: Regression coefficients (standardized) predicting transfer success based on semantic similarity of prompt embeddings.

| Model | Embedding | $N_{\text{suffix}}$ per prompt | Std. Coef. (Cosine Sim.) | $N$ | $R^2_m$ | $R^2_c$ |
|---|---|---|---|---|---|---|
| **Qwen** | Model | 100 | 0.651*** | 990,000 | 0.048 | 0.627 |
| | Indep. | 100 | 0.226*** | 990,000 | 0.006 | 0.608 |
| **Vicuna** | Model | 100 | 0.058*** | 990,000 | 0.000 | 0.785 |
| | Indep. | 100 | 0.041*** | 990,000 | 0.000 | 0.784 |
| **Llama 2** | Model | 100 | 0.622*** | 990,000 | 0.035 | 0.701 |
| | Indep. | 100 | 0.192*** | 990,000 | 0.004 | 0.687 |
| **Llama 3.2** | Model | 100 | 0.267*** | 990,000 | 0.008 | 0.643 |
| | Indep. | 100 | 0.159*** | 990,000 | 0.003 | 0.646 |

*Note:* Coefficients are standardized log-odds (mixed-effects logistic regression); $R^2_m$ = marginal $R^2$ (fixed effects only); $R^2_c$ = conditional $R^2$ (fixed + random effects). Random effects: (1|prompt) + (1|source_prompt/suffix). Stars denote statistical significance levels. * $p < 0.05$, ** $p < 0.01$, *** $p < 0.001$.

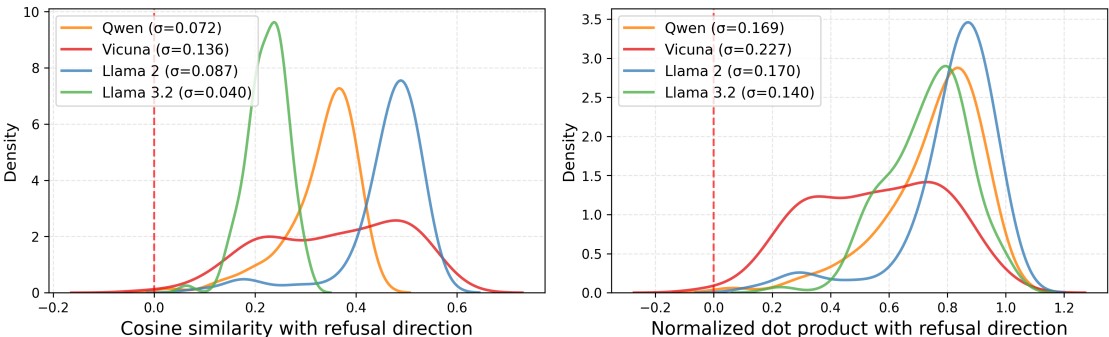

Figure 3: Distribution spread comparison of cosine similarity and (normalized) dot product activations with refusal direction across models.

semantic similarity contributes minimally to the fixed-effect component of explained variance, consistent with a small effect. The large gap between R²m and R²c further shows that most of the explained variance is attributable to prompt- and suffix-level random effects rather than semantic similarity.

### 5.3 Qualitative analysis of individual feature effects

We next *qualitatively* identify key geometric features that are correlated with jailbreak success, deferring a *quantitative* statistical analysis of these features until §5.4.

**Refusal connectivity.** In Figure 3, we plot the density of the cosine similarities and (normalized) dot products over the prompts with the refusal direction for the models we are considering. Vicuna has the largest spread, which could explain why the model is not capable of refusing some of the harmful questions without appending a suffix. The distributions are more concentrated for the other models, but there is still a reasonable spread in terms of the component along the refusal direction. In the statistical analysis, we will see how this variance in refusal connectivity is related to whether a suffix jailbreaks a prompt or not.

**Suffix push.** In Figure 4, we plot the distribution spread of prompts' refusal direction alignment given different suffix strengths for each model. This reveals several clear patterns. First, the average harmful prompt activation has the highest cosine similarity with the refusal direction (blue line). Furthermore, adding the three *least* successful suffixes (orange lines) only marginally reduces this cosine similarity, while adding the three *most* successful suffixes (green lines) significantly suppresses similarity with refusal.

**Orthogonal shift.** Figure 5 shows a positive relationship between suffix transferability (measured as the ASR over all tested prompts per suffix, see Definition 1) and both the orthogonal shift (Definition 6) and the

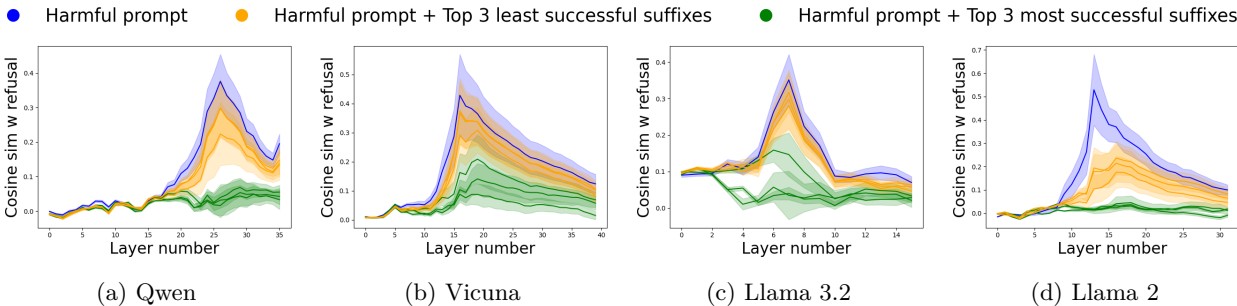

(a) Qwen     (b) Vicuna     (c) Llama 3.2     (d) Llama 2

Figure 4: Cross-layer suppression of refusal direction by most and least powerful suffixes for different models, figure based on Arditi et al. (2024). Activations are taken at the end-of-instruction token.

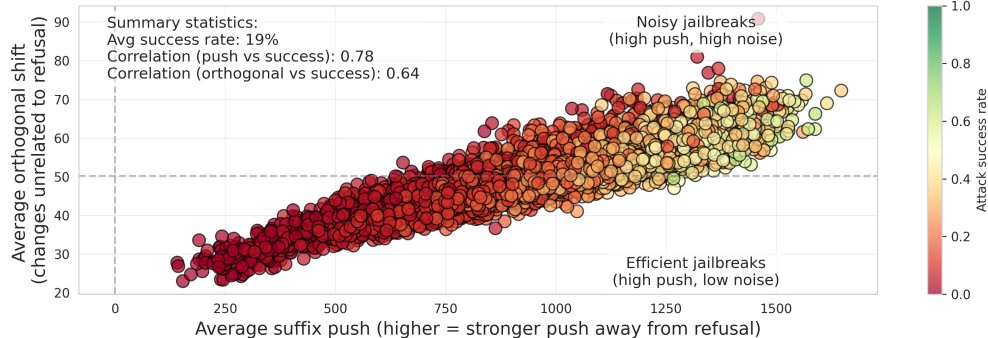

Figure 5: Qwen: Suffix effects on model representations (averaged across prompts for each suffix).

suffix push (Definition 5). This indicates that the likelihood of a successful transfer increases the more a suffix pushes away from refusal and also if it changes activations orthogonal to refusal. Similar patterns can be observed for the other models in Appendix C.

## 5.4 Quantitative analysis of feature effects

To quantitatively assess the impact of specific geometric features (defined in §3.2) on transfer, we formulate a logistic regression problem where, for each prompt-suffix pair $(i, j)$, we predict whether the suffix jailbreaks the prompt *solely* from the features of interest. This differs from the semantic similarity setup in §5.2, in that the covariates are prompt-suffix pairs, not prompt-prompt pairs.

**Statistical analysis setup.** For each prompt-suffix pair $(i, j)$, we define a binary target variable $y_{ij} \in \{0, 1\}$, where $y_{ij} = 1$ if suffix $j$ jailbreaks prompt $i$, and $y_{ij} = 0$ otherwise. We fit a separate generalized linear mixed model (GLMM) with a logistic link for each covariate $v \in \{s_i^{\text{base}}, \Delta_{ij}^{\text{push}}, \delta_{ij}^{\perp}\}$, where the feature vector takes the form $\mathbf{x}_{ij} := [1, v]$. Here $s_i^{\text{base}}$ is the refusal connectivity (Def. 4), $\Delta_{ij}^{\text{push}}$ is the suffix push (Def. 5), and $\delta_{ij}^{\perp}$ is the orthogonal shift (Def. 6). All predictors are standardized to have mean 0 and variance 1. Again, to account for the non-independence of observations sharing the same prompt or suffix, we include random intercepts for target prompt and for suffix nested within source prompt[2]. The resulting fixed-effect coefficients indicate the direction of each predictor's individual effect on transfer success. Note that we exclude pairs in which the suffix was originally optimized for a specific target prompt, as we are interested in suffix *transferability*, giving $N = 100 \times P(P-1) = 990{,}000$ observations per model.

**Results.** The results of the statistical analysis are presented in Table 2. Refusal connectivity has a negative and highly significant effect only for Vicuna, indicating that for this model refusal connectivity tends to

---

[2]in `lme4` notation (Bates et al., 2015): `(1 | prompt_index) + (1 | source_prompt_index / suffix_index)`

Table 2: Mixed-effects logistic regression coefficients (standardized) predicting transfer success based on single features.

| Variable | Qwen | Vicuna | Llama 2 | Llama 3.2 |
|---|---|---|---|---|
| Refusal connec. | 0.31 | −1.54*** | 0.08 | 0.07 |
| Suffix push | 2.59*** | 2.87*** | 3.56*** | 1.28*** |
| Orthogonal shift | 1.44*** | 0.08*** | 2.54*** | 1.06*** |
| Intercept | | | ✓ | |
| $N$ | 990,000 | 990,000 | 990,000 | 990,000 |

*Note:* Coefficients are standardized log-odds (mixed-effects logistic regression). Random effects: (1|prompt) + (1|source_prompt/suffix). Stars denote statistical significance levels. * $p < 0.05$, ** $p < 0.01$, *** $p < 0.001$.

dampen the likelihood of a successful suffix transfer to the prompt. In contrast, greater suffix push and also greater orthogonal shift are associated with a higher probability of transfer success for all models.

## 5.5 Analysis of joint effects for explaining adversarial transfer success

In the previous section, we studied the effects of how the individual factors of interest correlate with transfer. In this section, we combine them all in a *joint* statistical analysis aimed at determining how different features of the prompt and the suffix affect the likelihood that the suffix successfully jailbreaks the prompt. The joint analysis will allow us to probe the explanatory power of *all* features jointly, their relative effect magnitudes as well as the interdependencies between the features. The analyses focus on all features except semantic similarity given its different covariate setup. However, a repetition of the analyses including a related similarity-based feature is in Appendix C.

**Statistical analysis setup.** For each prompt-suffix pair $(i, j)$, we define a binary outcome variable $y_{ij} \in \{0, 1\}$, where $y_{ij} = 1$ if suffix $j$ successfully jailbreaks prompt $i$, and $y_{ij} = 0$ otherwise. To explain $y_{ij}$, we construct a feature vector $\mathbf{x}_{ij}$ capturing the properties of the prompt and the suffix we are interested in (defined in §3.3). Specifically, the feature vector $\mathbf{x}_{ij}$ is given by

$$\mathbf{x}_{ij} := [1, s_i^{\text{base}}, \Delta_{ij}^{\text{push}}, \delta_{ij}^{\perp}, s_i^{\text{base}} \cdot \Delta_{ij}^{\text{push}}, s_i^{\text{base}} \cdot \delta_{ij}^{\perp}, \Delta_{ij}^{\text{push}} \cdot \delta_{ij}^{\perp}]^{\top},$$

where $s_i^{\text{base}} \in \mathbb{R}$ is the refusal connectivity of the prompt (Definition 4), $\Delta_{ij}^{\text{push}} \in \mathbb{R}$ is the suffix push away from refusal (Definition 5), and $\delta_{ij}^{\perp} \in \mathbb{R}$ is the shift orthogonal to refusal (Definition 6). We standardize the coordinates of the feature vector so that they have mean 0 and variance 1. Note that the feature vector includes the individual factors as well as the pairwise products of these terms—this is because we will track the *main effects* due to these factors (i.e. the strength of the dependence of the $\{y_{ij}\}$ on these factors), as well as the *interaction effects* due to pairwise interactions between these factors (i.e. the strength of the pairwise dependence between these factors). This follows classical methodology in statistics (Hastie et al., 2009; Stock & Watson, 2015), according to which the coefficients we fit corresponding to the pairwise interaction effects capture how the influence of one variable changes depending on the value of another variable. This approach hence accounts for non-linear interactions between the main effects. We fit a generalized linear mixed model (GLMM) with a logistic link for this setup (i.e. we maximize the likelihood of the labels $\{y_{ij}\}$, such that for a choice of parameters $\boldsymbol{\beta} \in \mathbb{R}^7$, $\mathbb{P}(Y_{ij} = 1)$ is parametrized as $\exp(\boldsymbol{\beta}^{\top} \mathbf{x}_{ij})/[1 + \exp(\boldsymbol{\beta}^{\top} \mathbf{x}_{ij})]$). To account for the non-independence of observations sharing the same prompt or suffix, we include random intercepts for target prompt and for suffix nested within source prompt [3]. We exclude pairs in which the suffix was originally optimized for a specific target prompt, as we are interested in suffix *transferability*, giving $N = 100 \times P(P - 1) = 990,000$ observations per model.

**Intra-model transfer results.** The main effects (Table 3) are mostly in line with the single-factor results in §5.3 and §5.4. Higher refusal connectivity is associated with a decreased probability of transfer success;

---

[3] in lme4 notation (Bates et al., 2015): `(1 | prompt_index) + (1 | source_prompt_index / suffix_index)`

Table 3: Detailed mixed-effects logistic regression coefficients (standardized) with interaction effects predicting transfer success. Darker cell colors indicate larger effect sizes.

| Variable | Qwen | Vicuna | Llama 2 | Llama 3.2 | Llama 3.2 $\rightarrow$ Qwen | Qwen $\rightarrow$ Llama 3.2 |
|---|---|---|---|---|---|---|
| Refusal connec. | $-1.29^{***}$ | $-4.85^{***}$ | $-1.38^{***}$ | $-0.18$ | $-1.26^{***}$ | $-1.64$ |
| Suffix push | $3.05^{***}$ | $3.45^{***}$ | $3.61^{***}$ | $1.06^{***}$ | $1.34^{***}$ | $-0.99$ |
| Orthogonal shift | $0.29^{***}$ | $-0.78^{***}$ | $-0.28^{***}$ | $0.53^{***}$ | $0.53^{**}$ | $1.59$ |
| Interaction effects | | | | ✓ | | |
| Intercept | | | | ✓ | | |
| $N$ | 990,000 | 990,000 | 990,000 | 990,000 | 9,900 | 9,900 |
| $R_m^2$ | 0.688 | 0.495 | 0.592 | 0.194 | 0.354 | 0.040 |
| $R_c^2$ | 0.837 | 0.871 | 0.833 | 0.661 | 0.666 | 0.983 |

*Note:* Coefficients are standardized log-odds (mixed-effects logistic regression). Interactions are products of standardized predictors. $R_m^2$ = marginal $R^2$ (fixed effects only); $R_c^2$ = conditional $R^2$ (fixed + random effects). Random effects: (1|prompt) + (1|source_prompt/suffix). Stars denote statistical significance levels. $^{*}$ $p < 0.05$, $^{**}$ $p < 0.01$, $^{***}$ $p < 0.001$.

the effect is statistically significant for Qwen, Vicuna, and Llama 2. Greater suffix push is associated with higher probability of transfer success; the effect is statistically significant for all models. Similarly, the effect of the orthogonal shift is also statistically significant but mixed in direction, showing a positive impact on transfer success for Qwen and Llama 3.2 but a negative effect for Vicuna and Llama 2. Suffix push exhibits the largest effect for all models but Vicuna, for which refusal connectivity is most important. Orthogonal shift plays a less important role compared to the effect sizes of the other features. Note that all models include all pairwise interaction effects and an intercept. Given that all interaction effects are relatively small compared to the main effects, we focus on interpreting the main effects. Detailed results are shown in Appendix C.

**Inter-model transfer results.** The logistic regression for inter-model transfer (last two columns in Table 3) shows for Llama 3.2 to Qwen, that the main effects largely mirror the patterns observed in Qwen's intra-model analysis. For Qwen to Llama 3.2 no statistically significant effects were found. This is likely attributable to the overall very low success rate of transfers in this direction (as seen in Figure 2b), providing insufficient variance for the model to capture significant relationships.

**Takeaways.** In sum, these regression results point to broadly shared mechanisms influencing transfer success, with the suffix push being the most influential factor relative to other predictors (Table 3).

### 5.6 Interventional analysis

This section shows how our statistical insights can be used as interventions to improve attack success.

**Prompt rephrasing.** Our statistical analysis indicates that prompts more aligned with the refusal direction are harder to jailbreak, reducing suffix transfer. This suggests the following *interventional* experiment: testing whether rephrasing a prompt to be more or less aligned with refusal affects transfer. Using Vicuna, we generate 10 rephrases per prompt, compute their dot product with the refusal direction, and measure how dot product changes relate to ASR changes (see Appendix C for details). We expect a negative relationship as higher dot products should make it harder to transfer, lowering the ASR. Experiments with Qwen and Llama 3.2 confirm this (correlation coefficient Qwen: -0.08, $p < 0.05$, Llama 3.2: -0.18, $p < 0.05$). The significant relationship for both models suggests that our statistical insights can successfully guide intervention design.

**Altered GCG Loss.** Our statistical analysis indicates that suffixes inducing a larger *suffix push* or *orthogonal shift* are more likely to transfer. This suggests the following *interventional* experiment: modifying the GCG

loss to include regularizers favoring suffixes pushing away from or orthogonal to refusal. For these two settings, we evaluate Llama 3.2 with six non-zero regularization coefficients. We use 20 prompts—2 randomly taken from each of the 10 JailbreakBench categories—none of which jailbreak the model without a suffix.

Table 4: Results for altered GCG loss (Llama 3.2 model): Suffix Push.

| Coefficient | ASR | # jailbroken |
|---|---|---|
| 0 | 0.0138 | 552 |
| 0.00001 | 0.0177 | 709 |
| 0.0001 | 0.0189 | 757 |
| 0.001 | 0.0214 | 855 |
| 0.01 | 0.0176 | 706 |
| 0.1 | 0.0093 | 373 |
| 0.5 | 0.0101 | 406 |

Table 5: Results for altered GCG loss (Llama 3.2 model): Orthogonal Shift.

| Coefficient | ASR | # jailbroken |
|---|---|---|
| 0 | 0.0145 | 29 |
| 0.00001 | 0.0265 | 53 |
| 0.0001 | 0.0195 | 39 |
| 0.001 | 0.0175 | 35 |
| 0.01 | 0.0115 | 23 |
| 0.1 | 0.0020 | 4 |
| 0.5 | 0.0010 | 2 |

For the suffix push regularization term, we generate 100 suffixes for each of the 20 prompts, leading to 40,000 prompt/suffix pairs per coefficient. For the orthogonal shift regularization term, due to computational constraints, we generate 5 suffixes per prompt, leading to 2,000 prompt/suffix pairs per coefficient. We evaluate the ASR of the altered GCG algorithm using our jailbreak judge. We find that for both the suffix push and orthogonal shift regularization terms, the best coefficient is non-zero, corroborating our statistical analyses. Results are presented in Tables 4 and 5.

## 6 Conclusion

Our work identifies prompt- and suffix-specific factors that correlate strongly with successful suffix-based transfer. Through fine-grained statistical analysis, we characterize both the direction and strength of these effects, as well as their interplay. Among suffix-centric factors, the suffix push—the amount of shift away from the refusal direction—plays the strongest role across models. Among prompt-centric factors, the refusal connection—the alignment of a prompt embedding with the refusal direction—plays a strong role for certain models. Together, these factors contribute to a broader conceptual picture linking activations to the mechanisms underlying suffix-based transfer. Finally, through interventional experiments, we also demonstrate that these insights can be used to design stronger attacks and hope they can be used for developing stronger defenses.

## Acknowledgements

Part of this work was supported by the DAAD programme Konrad Zuse Schools of Excellence in Artificial Intelligence, sponsored by the German Federal Ministry of Education and Research (SB).

## Statement of broader impact

Our work contributes to a fundamental understanding of the vulnerabilities of LLMs. While we also show ways to make attacks more successful, we are convinced that our work will contribute to the development of technology that is safer to deploy and more aligned with societal benefits.

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

## A    Relegated definitions from Section 3.3

**Definition 7 (Optimal layer selection)** *Let $l^* \in \{1, 2, \ldots, L\}$ denote the optimal layer for extracting the refusal direction, where L is the total number of layers in the model. The optimal layer $l^*$ is selected as:*

$$l^* = \underset{l \in \{1,2,\ldots,L\}}{\arg\max} \; Effectiveness(\mathbf{v}_{refusal}^l) \tag{1}$$

*where $Effectiveness(\mathbf{v}_{refusal}^l)$ measures the success of the refusal direction at layer l in changing model behavior, following Arditi et al. (2024).*

For brevity, in the paper we drop the layer superscript $l^*$ when clear from context. All activations and refusal directions $\mathbf{v}_{\text{refusal}}$, $\mathbf{a}_i^{\text{base}}$, and $\mathbf{a}_{ij}^{\text{suffix}}$ are computed at the optimal layer $l^*$ unless explicitly stated otherwise.

## B    Relegated details for models and LLM judge reliability from Section 4

To test the reliability and accuracy of our LLM judge, we conducted an independent human evaluation on a stratified sample of 2,000 model responses drawn equally across all four models (Qwen2.5-3B-Instruct, Vicuna-13B-v1.5, LLaMA-2-7B-Chat, and LLaMA-3.2-1B-Instruct). For each model and prompt, we sampled up to 2 jailbroken and 3 non-jailbroken responses according to the judge's labels, falling back to 5 non-jailbroken samples for prompts where the judge found no jailbreaks — reflecting the naturally skewed label distribution in the data. This yielded 557 jailbroken and 1,443 non-jailbroken responses. Labels were verified by an independent research assistant, revealing an agreement rate of 98.65% with the automated judge, providing evidence against systematic labeling errors of our chosen jailbreak judge.

Table 6: Comparison of model selection

| Attribute | Qwen2.5-3B-Instruct | LLaMA-3.2-1B-Instruct | Vicuna-13B-v1.5 | LLaMA-2-7B-Chat |
|---|---|---|---|---|
| Alignment training | SFT, DPO, GRPO | SFT, DPO, RLHF | SFT | SFT, RLHF |
| Model size | 3B | 1B | 14B | 7B |
| # of generated suffixes | 10,000 | 10,000 | 10,000 | 10,000 |

## C    Additional qualitative and quantitative results relegated from Section 5

**Additional qualitative results for multiple random initializations**    Figure 6 shows the full prompt-by-seed jailbreak matrix for each model. These plots use the same filtering convention as Figure 1: the prompt axis includes only prompts that are not jailbroken without a suffix, excluding prompts that are already jailbroken in the no-suffix baseline.

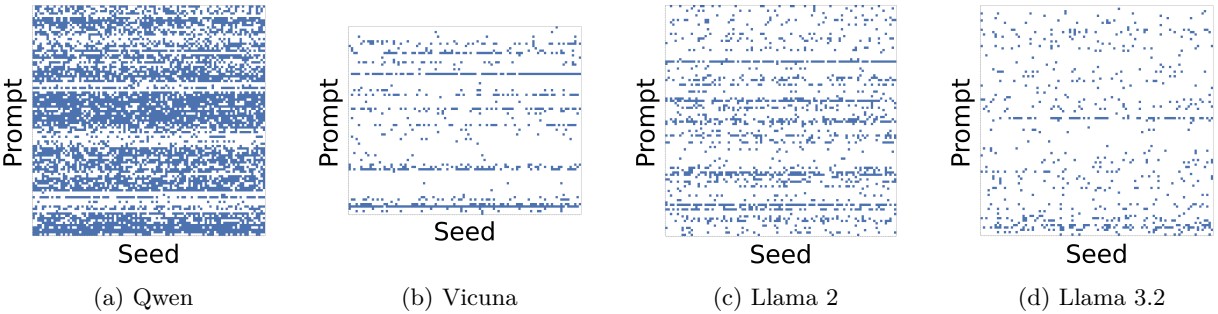

|   |   |   |   |
|---|---|---|---|
| (a) Qwen | (b) Vicuna | (c) Llama 2 | (d) Llama 3.2 |

Figure 6: Jailbreak success across multiple random initializations. Rows are restricted to prompts that are not jailbroken without a suffix. A cell at column $x$ and row $y$ is colored when the suffix generated for prompt $y$ using random seed $x$ successfully jailbreaks prompt $y$.

**Additional qualitative results for orthogonal shift** The following figures show the positive relationship between suffix transferability and both the orthogonal shift and suffix push features for Vicuna (Figure 7), Llama 2 (Figure 8), and Llama 3.2 (Figure 9). The main text includes a similar figure for Qwen (see Figure 5). For all models we observe a similar trend of higher suffix push and higher orthogonal shift being correlated with suffix transferability, albeit with less strong signal for the Llama models. This is because there are less examples of successful transfers in general.

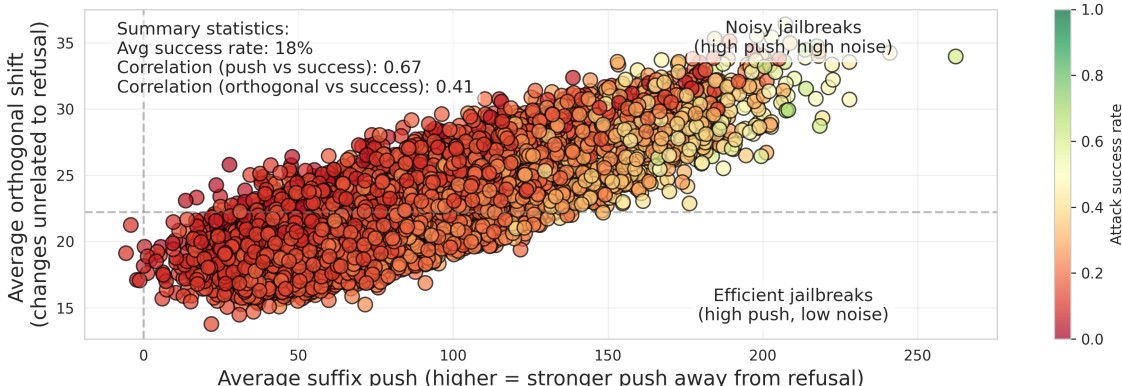

Figure 7: Suffix orthogonal shift and push effects on model representations (averaged across harmful prompts for each suffix ID) for Vicuna.

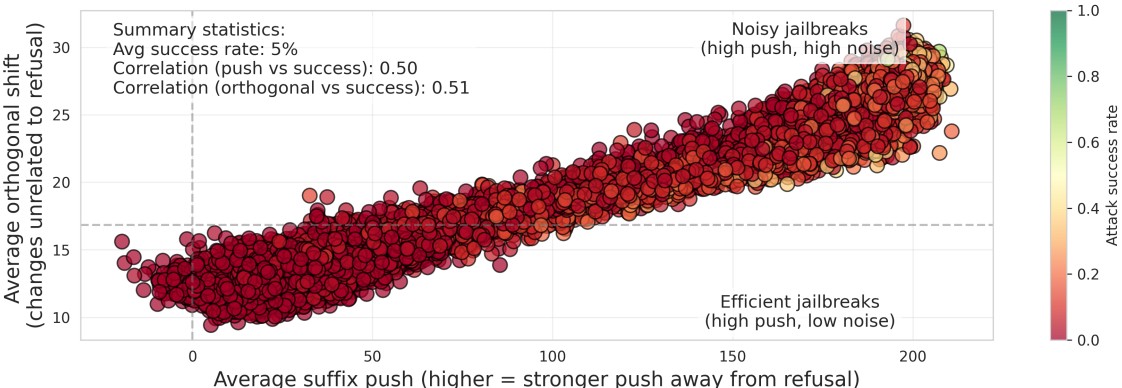

Figure 8: Suffix orthogonal shift and push effects on model representations (averaged across harmful prompts for each suffix ID) for Llama 2.

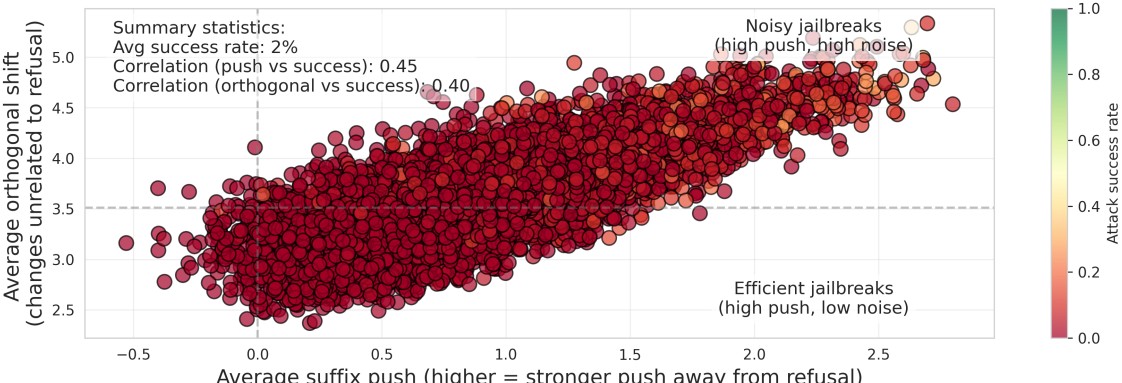

Figure 9: Suffix orthogonal shift and push effects on model representations (averaged across harmful prompts for each suffix ID) for Llama 3.2.

Table 7: Detailed mixed-effects logistic regression coefficients (standardized) with interaction effects predicting transfer success.

| Variable | Qwen | Vicuna | Llama 2 | Llama 3.2 | Llama 3.2 → Qwen | Qwen → Llama 3.2 |
|---|---|---|---|---|---|---|
| Refusal connec. | $-1.29^{***}$ | $-4.85^{***}$ | $-1.38^{***}$ | $-0.18$ | $-1.26^{***}$ | $-1.64$ |
| Suffix push | $3.05^{***}$ | $3.45^{***}$ | $3.61^{***}$ | $1.06^{***}$ | $1.34^{***}$ | $-0.99$ |
| Orthogonal shift | $0.29^{***}$ | $-0.78^{***}$ | $-0.28^{***}$ | $0.53^{***}$ | $0.53^{**}$ | $1.59$ |
| Refusal connec. $\times$ Suffix push | $0.39^{***}$ | $0.51^{***}$ | $0.09^{**}$ | $-0.19^{***}$ | $0.03$ | $-0.86$ |
| Refusal connec. $\times$ Orthogonal shift | $0.69^{***}$ | $-0.25^{***}$ | $-0.29^{***}$ | $0.01$ | $0.19$ | $1.36$ |
| Suffix push $\times$ Orthogonal shift | $-0.91^{***}$ | $-0.06^{***}$ | $0.40^{***}$ | $0.02^{*}$ | $-0.07$ | $0.52$ |
| Intercept | $-3.51^{***}$ | $-4.93^{***}$ | $-7.46^{***}$ | $-6.56^{***}$ | $-6.47^{***}$ | $-23.25^{***}$ |
| $N$ | 990,000 | 990,000 | 990,000 | 990,000 | 9,900 | 9,900 |
| $R_m^2$ | 0.688 | 0.495 | 0.592 | 0.194 | 0.354 | 0.040 |
| $R_c^2$ | 0.837 | 0.871 | 0.833 | 0.661 | 0.666 | 0.983 |

*Note:* Coefficients are standardized log-odds (mixed-effects logistic regression). Interactions are products of standardized predictors. $R_m^2$ = marginal $R^2$ (fixed effects only); $R_c^2$ = conditional $R^2$ (fixed + random effects). Random effects: (1|prompt) + (1|source_prompt/suffix). Stars denote statistical significance levels. $^{*}$ $p < 0.05$, $^{**}$ $p < 0.01$, $^{***}$ $p < 0.001$.

**Additional results for the analysis of joint effects in Section 5.5** In Section 5.5, we calculate the joint effect of our features of interest in a logistic regression analysis. While Table 3 in the main text focuses on the main effects, Table 7 details the regression coefficients for all interaction effects and the intercept.

The interaction effects are mostly substantially smaller than the main effects—so one should be careful not to read too much into the specific sign patterns.

Table 8 displays the same logistic regression model with an added coefficient for semantic similarity. Semantic similarity is calculated as the similarity of embeddings between two prompts (as described in Definition 3). In this regression analysis, we use the semantic similarity based on model internal activations on the last instruction token at the layer where the refusal direction is extracted.

We observe that semantic similarity has a positive and highly statistically significant effect on transfer success (except for Vicuna where the effect is negative), which means that if two prompts have high similarity in activations, their suffixes are more likely to successfully transfer. However, compared to the size of the coefficients for suffix push and refusal connectivity, the influence is relatively small. Again, the interaction effects are small in size compared to the main effects.

Table 8: Mixed-effects logistic regression coefficients (standardized) predicting transfer success (intra-model) including semantic similarity based on model internal embeddings.

| Variable | Qwen | Vicuna | Llama 2 | Llama 3.2 |
|---|---|---|---|---|
| Semantic similarity (model) | 0.25*** | -0.05*** | 0.62*** | 0.25*** |
| Refusal connectivity | -1.46*** | -4.83*** | -1.47*** | -0.26 |
| Suffix push | 3.04*** | 3.45*** | 3.62*** | 1.03*** |
| Orthogonal shift | 0.29*** | -0.78*** | -0.28*** | 0.57*** |
| Refusal connec. $\times$ Suffix push | 0.36*** | 0.47*** | 0.25*** | -0.20*** |
| Refusal connec. $\times$ Orthogonal shift | 0.76*** | -0.27*** | -0.30*** | 0.05** |
| Suffix push $\times$ Orthogonal shift | -0.90*** | 0.06*** | 0.38*** | 0.02* |
| Refusal connec. $\times$ Sem. similarity | 0.00 | -0.13*** | 0.22*** | -0.01 |
| Suffix push $\times$ Sem. similarity | -0.01 | 0.08*** | -0.34*** | 0.02 |
| Orthogonal shift $\times$ Sem. similarity | -0.09*** | 0.03*** | 0.01 | -0.08*** |
| (Intercept) | -3.50*** | -4.86*** | -7.59*** | -6.58*** |
| N | 990,000 | 990,000 | 990,000 | 990,000 |
| $R^2 m$ | 0.693 | 0.493 | 0.608 | 0.205 |
| $R^2 c$ | 0.839 | 0.870 | 0.846 | 0.659 |

*Note:* Coefficients are standardized log-odds (mixed-effects logistic regression). Interactions are products of standardized predictors. $R^2_m$ = marginal $R^2$ (fixed effects only); $R^2_c$ = conditional $R^2$ (fixed + random effects). Random effects: (1|prompt) + (1|source_prompt/suffix). Stars denote statistical significance levels. * $p < 0.05$, ** $p < 0.01$, *** $p < 0.001$.

**Prompt rephrasing instructions**  In Section 5.6 we use Vicuna to rephrase the harmful questions without changing their meaning. To generate diverse rephrases of jailbreak prompts, we use the following template prompt:

```
In the following you will get a prompt.  Please generate {num_rephrases}
rephrases of this prompt that have the same meaning and will lead to a
similar answer.  You can add paraphrases that contain setting the prompt
into a hypothetical scenario, too, like "hypothetically speaking, ...".
Each of the {num_rephrases} rephrases should be unique.

Original prompt:  {prompt}

Please provide exactly {num_rephrases} rephrases, each on a new line,
numbered 1-{num_rephrases}:
```

In this template:

- {num_rephrases} is replaced with the desired number of rephrases to generate

- {prompt} is replaced with the original jailbreak prompt to be rephrased

This systematic approach ensures consistent generation of semantically equivalent variants while maintaining the adversarial intent of the original prompts.

