# OpenReview forum: "Towards Understanding the Transferability of Adversarial Suffixes in Large Language Models"
_TMLR — Accepted by TMLR_

### Review · Reviewer_Xiyx · 2026-02-05

**Summary Of Contributions:**

This work empirically investigates the transferability of evasion attacks / jailbreaks on LLMs. Specifically, it focuses on adversarial suffixes and to what extend different factors related to refusal direction (Arditi et al., 2024) are predictive of transferability.
Concretly, these factors quantify: How aligned a harmful prompt (without adversarial suffix) is with the refusal direction in embedding space, how much a suffix pushes a prompt in the opposite direction, and how much it modifies the prompt in an orthogonal direction.

After formally defining these three quantities, the authors conduct experiments with four models, $P=100$ harmful prompts and either $K=1$ or $K=100$ adversarial suffixes per prompt and model.

The authors first demonstrate that, qualitatively, different suffixes appear to have varying levels of transferability across prompts for the same model and across different models.
The relation between transferability and the introduced factors is then quantified using different forms of regression analysis and the associated significance levels.

Finally, the authors demonstrate that these insights can be exploited for adversarial attacks or defenses by (1) rephrasing prompts to be more or less aligned with the refusal direction or (2) altering the attack objective to favor perturbations orthogonal/antiparallel to the refusal direction.

## Strengths
* Even though novelty is not a main decision criterion, investigating transferability for LLMs from a mechanistic perspective is a new and interesting direction
* Related work on transferabiliy for both LLMs and non-LLM is discussed in sufficient detail
* The choices of potential factors governing transferability seem natural and are well-motivated
* The paper is well-written and easy to follow (except for some missing information in the experimental setup, see below). In particular the experimental section (qualitative --> quantitative --> intervention) flows very nicely.
* The authors actually evaluate the statistical significance of their findings, which is good and uncommon for ML papers

## Weaknesses
* There appear to be multiple possible sources of error in the reported significance levels, see ``correctness'' below
* Experiments are only conducted with 100 prompts, but this limitation is shared by all papers using the JailbreakBench benchmark
* The used number of suffixes varies across models and forces the authors to use different types of regression coefficients, which makes some results somewhat hard to compuare.
* In Section 5.1 and particularly Figs.1-2, it is not sufficiently clear how exactly transferability is evaluated. It would be great if the authors could explicitly state that each point (x,y) in each subfigure corresponds to an adversarial suffix construced on prompt y and applied to prompt x.

**Additional Comments:**

This work is largely based on the assumption from (Arditi et al., 2024) that there exists a single direction of refusal.
However, a follow-up work from last year (Wollschläger et al., 2025) demonstrates that there are multiple refusal direction that form a subspace, and calls into question the use of orthogonal manipulation to independently control refusal.

Obviously, dozens of papers like this appear every day, none of them actually formally prove their claims, and no one can be expected to read even a fraction of them. So **your response to this question will not influence my decision and I don't think that the other paper  needs to be cited**.

But could you briefly comment on whether the chosen features in your Section 3.2 actually make sense if we assume that there are multiple refusal directions?


[1] Wollschläger et al. The Geometry of Refusal in Large Language Models: Concept Cones and Representational Independence. ICML 2025. https://arxiv.org/abs/2502.17420

**Audience:**

Yes

**Audience Explanation:**

Yes. Adversarial robustness of LLMs is still one of the most actively researched topics in trustworthy ML. Similar mechanistic works that empirically measure the behavior of LLMs in adversarial settings regularly appear in NeurIPS/ICLR/ICML/TMLR.

**Claims And Evidence:**

No

**Claims Explanation:**

As mentioned above, there are some inherent limitations in the experimental setup (e.g., only considering 100 prompts), but these are shared across various papers in the community.

Furthermore, the experimental results for larger models would be more convincing of the same number of suffixes was used everywhere.

While it is commendable that the authors actually perform statistical tests for their hypotheses, I believe there are multiple errors that should be corrected or at least discussed. With the current experimental setup, **I do not believe that any of the reported significance levels are valid**.

## Table. 1
There are $P=100$ prompts and $K=1$ or $K=100$ suffixes per prompt. The coefficients are derived from regression on feature-target pairs $x_{p,p'}, y_{p,p'}$. Here, $p$ and $p'$ are two different prompts and $y_{p,p'}$ is derived from a reduction over the ``K'' dimension (average transfer success).
Therefore, we should have $N=10000$. It is not clear how the authors arrive at $N=10^6$ or $N=10^5$ and why $N$ should differ between $K=1$ and $K=100$.

The used statistical tests assume $N = P^2$ samples to be drawn i.i.d. from some distribution over prompt pairs. However, the actual experiment (1) draws $P=100$ prompts, and (2) generate $N=P^2$ pairs thereof, which is a different distribution. It is not clear if and how the authors adjust for this mismatch.

## Table. 2
Here, regression coefficients are computed for prompt-suffix pairs. It is not clear how we arrive at $N=8 \cdot 10^3$ or $N=8 \cdot 10^5$ pairs.

The statistical test assumes that we draw $N$ prompt-suffix pairs i.i.d. from some distribution. However, the actual experiment (1) draws $P$ suffixes and (2), for each prompt, draws $K$ suffixes. Again, this introduces some correlations that violate the usual independence assumption of these tests. This does not seem to be accounted for.

## Table 3
Same as Table 2.

**Requested Changes:**

* Conduct all experiments with $K=100$ suffixes. There should be sufficient time to do this in the months leading up to the final submission. I do not expect this to be completed within the 2-week rebuttal period.
* Please address all of the issues with the statistical tests listed above
* Use more descriptive captions for Figs. 1-3 that explain correspondences between prompt-suffix pairs across models.

---

> ### Author Response · Authors · 2026-04-10
>
> Thanks a lot for your helpful review and for acknowledging the novelty of our work. We address your concerns in the following:
>
> **Weakness 1 “only 100 prompts”:**\
> The choice to use 100 prompts is standard in this literature as JailbreakBench is one of the most commonly used benchmarks in the field of jailbreaking. The use of this standard benchmark improves reproducibility; others can use existing codebases to rerun and reproduce our experiments, which would be more difficult if we were to use less standard datasets.
>
> **Weakness 2 “different number of suffixes and different statistical models”:**\
> We re-run the semantic similarity analysis with a different setup that aligns with the setup of the other regressions, making the (standardized) regression coefficients more comparable. For details please see our responses to the concerns below.
>
> We commit to generating more suffixes for the two bigger models too but as you mentioned need more time to achieve this beyond the two-week rebuttal period.
>
> **Weakness 3 “clarifying section 5.1 transferability”:**\
> Thank you for pointing out this potential point of confusion. Exactly as you have stated, we will revise the paper to make explicit that in figures 1 and 2, a point (x,y) corresponds to an adversarial suffix constructed on prompt y and applied to prompt x.
>
> **Concerns with statistical tests:**
>
> **Table 1:**\
> We revised the analysis with a unified setup that addresses both concerns:
>
> *What changed.* In the original analysis (Table 1), the two suffix conditions used structurally different models: for K = 100, we first averaged the binary jailbreak outcome over all 100 suffixes per source prompt p, yielding a continuous y_{pp'} ∈ [0,1] per prompt pair, and fit a linear regression; for K = 1, we fit an ordinal logistic regression on the quasi-discrete outcome y_{pp'} ∈ {0,0.5, 1}. In the new setup, we *no longer* average over suffixes. Every observation is now a single (source prompt p, suffix s, target prompt p') triple with a binary outcome indicating whether the target prompt p’ was jailbroken. Now, both conditions also use the same logistic regression framework, predicting how semantic similarity of the pair p,p’ influences transferability of suffix s on p’. Note that we exclude same-prompt pairs (p = p'). This gives:
>
> - **K = 1**: N = P(P−1) = 9,900
> - **K = 100**: N = 100 × P(P−1) = 990,000
>
> *Accounting for non-independence.* As you rightly point out, this setup means that many observations in the dataset share correlational structure as there are observations that share the same p, p’ or suffix.  We fit a GLMM with random intercepts to address the non-i.i.d. structure:
>
> - **K = 1**: random intercepts for *target prompt id* and *source prompt id*, which is `(1 | target_prompt_id) + (1 | source_prompt_id)` in the lme4 notation (R package). Since each source prompt has exactly one suffix, suffix identity is perfectly confounded with source prompt identity and a separate suffix random effect is not identified, which is why it is not included separately.
> - **K = 100**: random intercepts for target prompt id and for suffix id nested within source prompt id — written `(1 | target_prompt_id) + (1 | source_prompt_id / suffix_id)` in lme4 notation (R package). The nesting captures the additional variance shared by the 100 suffixes co-optimised on the same source prompt.
>
> *Results.* The updated results are consistent with Table 1 in direction and largely in significance — though the coefficients for Llama 2 lose statistical significance.
>
>
> **Table 2 and 3:**\
> In Tables 2 and 3, the original N values of ~8×10³ and ~8×10⁵ reflected a train-test split that we have now removed. The unit of observation here is a (target prompt, suffix) pair, where we exclude pairs in which the suffix was originally optimised for that target prompt, as we are interested in *transferability*. This gives N = P(P−1) = 9,900 for K=1 and N = 100 × P(P−1) = 990,000 for K=100; consistent with Table 1.
> To address the non-independence concern, we apply the same GLMM approach as in Table 1:
> - **K = 1**: random intercepts for *target prompt id* and *source prompt id*, which is `(1 | target_prompt_id) + (1 | source_prompt_id)` in the lme4 notation (R package).
> - **K = 100**: random intercepts for target prompt id and for suffix id nested within source prompt id — written `(1 | target_prompt_id) + (1 | source_prompt_id / suffix_id)` in lme4 notation (R package).
>
> Again, the updated results are consistent with the original findings in direction and largely in significance, though some coefficients lose statistical significance. Our overall interpretations remain unchanged. We will update the tables and discussions accordingly (including the to-be generated results for the 100 suffixes on larger models).

---

> > ### Comment · Reviewer_Xiyx · 2026-04-10
> >
> > Thank you. Assuming the promised experiments are included (especially more suffixes for large models), this resolves the concerns I raised in my original review.

---

### Review · Reviewer_mdR2 · 2026-02-14

**Summary Of Contributions:**

The paper explains why adversarial suffix jailbreaks could transfer across prompts and sometimes models: transfer is more likely when the base prompt has low refusal connectivity, and when the suffix induces both a strong push away from refusal and a large orthogonal shift in representation space, while prompt semantic similarity is only a weak predictor by comparison. The paper supports this with large-scale experiments and uses the findings to motivate small interventions like prompt rephrasing and modified GCG objectives that can improve attack success.

Key strengths:
- the paper defines concrete quantities and tests them systematically.
- it generates many suffixes per prompt via different random seeds, which helps reduce “one lucky suffix” anecdotes and makes the correlations more credible.
- it attempts to translate analysis into actionable interventions.

Key weaknesses:
- since much of the story hinges on a single “refusal direction,” the paper’s conclusions inherit sensitivity to how that direction is computed and which layer is chosen; some details are deferred/brief.
- uneven methodology across models: for larger models, are suffixes generated in the same multi-seed way?
- the modified-loss experiments are demonstrated on limited prompts/models, so generality is suggestive rather than fully established.

**Audience:**

Yes

**Audience Explanation:**

It studies LLMs which is an active / popular field.

**Claims And Evidence:**

Yes

**Claims Explanation:**

Most claims are well supported. I listed below my concerns:

- Independence assumptions likely violated in regressions. The logistic regressions treat prompt-suffix pairs as independent observations (Secs 5.4, 5.5), but the design has strong clustering (many datapoints share the same prompt and/or suffix). Without further clarification, the assumption is not fully convincing.

- Cross-model evidence is methodologically uneven. For Qwen/Llama-3.2 the paper generates 100 suffixes per prompt; for Vicuna/Llama-2 it uses one archived suffix per prompt due to compute limits. That means “across models” generality is supported by unequal-quality evidence.

- Reliance on a single LLM judge without additional validation. The evaluation uses a Llama-3-70B judge per JailbreakBench recommendations, but the paper doesn’t provide robustness checks (e.g., alternate judges, spot checks, disagreement analysis). If the judge has systematic bias, that bias becomes the “evidence.”

**Requested Changes:**

Address the weaknesses and my concerns above.

---

> ### Author Response · Authors · 2026-04-10
>
> Thanks a lot for this helpful review and for appreciating our thorough statistical analysis! In the following, we will address the weaknesses and concerns you raised:
>
> **Weakness 1 “refusal direction”:**\
> We base the derivation of the refusal direction on the seminal paper [1], who did extensive validations that the refusal direction is meaningful and which layer should best be selected. It would of course be valuable future work to investigate how well our conclusions transfer to different, less widely used extraction methods—though carrying out such a study in a systematic way would be nontrivial and, in our view, substantial enough to warrant a separate paper.
> The details of the procedure are outlined in the appendix (Section A). If the review believes it would be helpful, we can move these details from the appendix to the main body of the paper and elaborate in more detail.
>
> **Weakness 2 “uneven methodology across models”:**\
> In our submission, we had two models with multi-seed suffixes and two models with only one seed per prompt, resulting in either 100 suffixes or 1 suffix per prompt. We are extending the experiments to also generate 100 suffixes per prompt for the two bigger models. However, this takes a lot of computational time, which is why we expect to get the results after the rebuttal period but commit to including those results in the camera-ready version.
>
> **Weakness 3 “modified-loss experiments on limited prompts”:**\
> We will extend the analysis here too but will need more time than the rebuttal period for the experiments to complete.
>
> **Concern 1 “independence assumption”:**\
> We repeated the analyses with a mixed-effects design that includes random effects for the prompts and the suffixes plus the nested suffix structure in the multi-seed case (see also our responses to the concerns of reviewer Xiyx). The results remain mainly the same and will be included in the revision of the submission.
>
> **Concern 2 “generalizability of cross model evidence”:**\
> As mentioned in our answer to weakness 2, we will expand the analyses to have 100 suffixes for all models.
>
> **Concern 3 “reliance on single LLM as judge”:**\
> We use the Llama-3-70B jailbreak judge following JailbreakBench [2], which validated this choice against majority-vote labels from three expert annotators across 300 prompt-response pairs. Among six candidate judges, Llama-3-70B achieved the highest human agreement (90.7%) and lowest false negative rate (5.5%), while deliberately erring conservative on false positives (11.6%) to avoid misclassifying benign responses as jailbroken.
>
> Nonetheless, to address the reviewer’s concern that systematic judge bias could propagate into our conclusions, we conducted an independent human evaluation on a stratified sample of 2,000 model responses drawn equally across all four models (Qwen2.5-3B-Instruct, Vicuna-13B, LLaMA-2-7B-Chat, and LLaMA-3.2-1B-Instruct). For each model and prompt, we sampled up to 2 jailbroken and 3 non-jailbroken responses according to the judge's labels, falling back to 5 non-jailbroken samples for prompts where the judge found no jailbreaks — reflecting the naturally skewed label distribution in the data. This yielded 557 jailbroken and 1,443 non-jailbroken responses. Labels were verified by an independent research assistant, revealing an agreement rate of 98.65% with the automated judge, providing evidence against systematic labeling errors of our chosen jailbreak judge.
>
>
> [1] Arditi, Andy, et al. "Refusal in language models is mediated by a single direction." Advances in Neural Information Processing Systems 37 (2024): 136037-136083.
>
> [2] Chao, Patrick, et al. "Jailbreakbench: An open robustness benchmark for jailbreaking large language models." Advances in Neural Information Processing Systems 37 (2024): 55005-55029.

---

> > ### Comment · Reviewer_mdR2 · 2026-04-18
> >
> > Thanks for the authors' rebuttal. Almost all my concerns were addressed. My further comment: the authors promised to provide more experimental results in the camera-ready version, becasue of limit of time. Would they please provide timeline and plan.

---

> > > ### Author Response · Authors · 2026-04-28
> > >
> > > Sure, we are already working on the extensions. Calculated conservatively given the current compute we have access to, our plan and timeline looks like the following:
> > >
> > > Generating suffixes: 2 weeks
> > >
> > > Generating answers and judge evals: 1-2 weeks
> > >
> > > Extracting activations: 2 days
> > >
> > > Running statistical analysis: one day

---

> > > > ### Comment · Reviewer_mdR2 · 2026-04-28
> > > >
> > > > Thanks, sounds good. Look forward to the results.

---

### Review · Reviewer_A9BS · 2026-04-03

**Summary Of Contributions:**

### Summary

This paper studies the transferability of adversarial suffixes in LLMs, a clearly defined and practically important problem. It approaches this question through the lens of the refusal direction, and analyzes several geometric factors in representation space that correlate with transfer success, such as prompt alignment with refusal, suffix push away from refusal, and orthogonal shift. The paper further supports this framing with large-scale statistical analysis and interventional experiments, providing a more mechanistic understanding of adversarial suffix transferability.

---
### Strengths
1. The paper studies an important and well-scoped problem, namely the transferability of adversarial suffixes in LLMs.
2. The analysis is built around a coherent mechanistic perspective centered on the refusal direction, providing a unified lens for studying both prompt susceptibility and suffix effectiveness.
3. The paper provides a fine-grained analysis of the factors associated with transfer success, including both individual and joint effects.
4. The interventional experiments are a valuable addition

---
### Weaknesses
1. It remains somewhat unclear how robust the main conclusions are to the specific method used to extract the refusal direction. The paper builds almost the entire analysis around a single refusal direction framework, but it does not sufficiently examine whether the conclusions remain stable under alternative extraction methods. This is an important concern because recent work has argued that refusal may not be governed by a single global direction, but instead by multiple directions or richer geometric structures such as [1]
2. The data setup is highly imbalanced between small and large models. For small models, the paper uses 100 suffixes per prompt, whereas for larger models it uses only a single suffix per prompt from the JailbreakBench archive. This makes cross-model comparisons substantially less clean
3. The paper evaluates jailbreak success using a Llama-3-Instruct-70B judge. While this is a reasonable open-model choice, the paper does not sufficiently justify why this judge is preferred over more advanced frontier models or stronger closed-model evaluators.
4. The paper show an interesting observation that some suffixes fail on the prompt they were optimized for, yet succeed on other prompts within the same model. This is an interesting phenomenon, but the paper does not go much further in explaining why it happens.
5. The intervention with altered GCG loss has a good motivation, but the empirical gains are fairly limited.


[1] Wollschläger, Tom, et al. "The Geometry of Refusal in Large Language Models: Concept Cones and Representational Independence." Forty-second International Conference on Machine Learning.

**Audience:**

Yes

**Audience Explanation:**

Yes. The paper studies an important safety problem in LLMs and offers a systematic analysis of adversarial suffix transferability, which should be of interest to readers working on LLM safety and mechanistic interpretability.

**Broader Impact Concerns:**

No ethical concerns.

**Claims And Evidence:**

Yes

**Claims Explanation:**

The paper provides fairly clear empirical support for its claims The central findings about the roles of refusal connectivity, suffix push, and orthogonal shift are supported by consistent regression results across multiple models and settings.

Addressing Weaknesses 1, 2, and 3 would further strengthen the evidence and improve the overall credibility.

**Requested Changes:**

Addressing Weaknesses 1, 2, and 3 would further strengthen the evidence and improve the overall credibility.

Weaknesses 4 and 5 are more in the category of nice-to-have improvements rather than core concerns.

---

> ### Author Response · Authors · 2026-04-10
>
> Thanks a lot for your thorough feedback and for appreciating our thorough statistical analysis! In the following we will address the weaknesses you raised.
>
> **Weakness 1**\
> The method we use to extract a refusal direction is the most standard and widely used in the literature. While alternative methods have been proposed, including the one mentioned by the reviewer, none are comparably established at this point. We agree that it would be valuable future work to investigate how well our conclusions transfer across these alternative extraction methods. That said, carrying out such a study in a systematic way would be nontrivial and, in our view, substantial enough to warrant a separate paper.
>
> **Weakness 2** \
> Please see our response to Weakness 2 to reviewer mdR2.
>
> **Weakness 3**\
> Please see our response to Weakness 3 to reviewer mdR2. We will add this discussion and the results of the human evaluation to the paper.
>
> **Weakness 4**\
> The phenomenon is indeed very interesting and surprising. We believe it points to a richer, non-trivial interaction between prompt structure, optimization, and model-specific refusal mechanisms. Unfortunately we do not have a deeper understanding currently.
>
> **Weakness 5**\
> Our modification to GCG was intended primarily as a proof of concept: to show that an intervention directly motivated by our statistical analysis can improve attack success. We suspect that additional engineering effort could further amplify these empirical gains.

---

### Decision · Action_Editor_VwJt · 2026-05-03

**Recommendation:** Accept with minor revision

**Additional Comments:**

The acceptance is conditional upon the authors fulfilling their commitment to complete the experiments in their rebuttal time, including

- Extend the evaluation to include 100 suffixes per prompt for the larger models.

- Integrate the human validation results into the final manuscript.

- fix clarity issues.

**Audience:**

Yes

**Audience Explanation:**

This work is highly relevant to the machine learning safety and mechanistic interpretability communities.

**Claims And Evidence:**

Yes

**Claims Explanation:**

The paper provides a principled mechanistic analysis of adversarial suffix transferability. The claims are supported by rigorous statistical analysis and validated via experiments.